# Sulfatide Binds to Influenza B Virus and Enhances Viral Replication

**DOI:** 10.3390/v17040530

**Published:** 2025-04-05

**Authors:** Yuuki Kurebayashi, Yoshiki Wakabayashi, Tadanobu Takahashi, Keiko Sakakibara, Shunsaku Takahashi, Akira Minami, Takashi Suzuki, Hideyuki Takeuchi

**Affiliations:** 1Department of Biochemistry, School of Pharmaceutical Sciences, University of Shizuoka, 52-1 Yada, Suruga-ku, Shizuoka 422-8526, Shizuoka, Japan; kurebayashi@u-shizuoka-ken.ac.jp (Y.K.); takahasi@u-shizuoka-ken.ac.jp (T.T.);; 2Department of Functional Morphology, Faculty of Pharmacy, Juntendo University, 6-8-1 Hinode, Urayasu 279-0013, Chiba, Japan; a.minami.da@juntendo.ac.jp

**Keywords:** antivirus therapeutics, hemagglutinin, influenza B virus, sulfatide, viral replication

## Abstract

Seasonal influenza epidemics caused by influenza A viruses (IAV) and influenza B viruses (IBV) pose a substantial public health burden. Despite the significant impact of IBV, its restricted host range and the absence of documented pandemics have resulted in limited research attention relative to IAV. Understanding the viral infection mechanisms of both IAV and IBV is crucial for controlling seasonal epidemics. Previously, we demonstrated that 3′-*O*-sulfated galactosylceramide sulfatide binds to IAV and enhances viral replication, a finding with potential therapeutic implications. However, the role sulfatide plays in other influenza virus infections, including those caused by IBV, remains unknown. Accordingly, in this paper, we investigate the function of sulfatide during IBV infection. We demonstrate that sulfatide binds to IBV hemagglutinin (HA), and that sulfatide overexpression significantly enhances IBV replication, whereas treatment with sulfatase or an anti-sulfatide antibody markedly suppressed IBV replication. Moreover, further tests involving the inhibition of sulfatide biosynthesis resulted in the suppression of viral replication with impaired nuclear export of viral ribonucleoproteins (vRNPs). These findings establish that sulfatide is a critical regulator of IBV replication, which parallels its role in IAV infection, and suggest that targeting sulfatide-virus interactions can lead to broad-spectrum therapeutic strategies against influenza virus.

## 1. Introduction

Influenza viruses are classified into four types, A, B, C, and D, based on their internal antigens. Among these, the influenza B virus (IBV) primarily infects humans and causes acute respiratory illness [1].

Seasonal influenza epidemics are driven by both influenza A virus (IAV) and IBV due to antigenic drift in viral surface proteins [2]. Unlike IAV, which poses a pandemic risk due to its broader host range and higher level of interspecies transmission, IBV exhibits a narrow host range restricted to humans, seals, and dogs [3,4]. Even though IBV is not known to cause pandemics, it remains a significant public health concern and can cause seasonal epidemics. Vaccination is the primary strategy for preventing influenza, and current seasonal vaccines include components that target IBV. However, vaccine effectiveness varies by season and is influenced by antigenic drift within the IBV genome [5]. The structural differences between IAV and IBV also affect the effectiveness of antiviral treatments. For example, while neuraminidase inhibitors such as oseltamivir, zanamivir, and peramivir demonstrate clinical efficacy against both IAV and IBV [6], M2 ion channel inhibitors are effective only against IAV [7]. Therefore, controlling seasonal influenza epidemics requires countermeasures against both IAV and IBV.

Similar to IAV, the IBV genome is composed of eight segments, with both viruses sharing key structural components [8]. In addition, both viruses rely on similar entry processes, which involve the binding of their surface glycoproteins to host cell receptors, followed by endocytosis and the eventual release of viral RNA into the host cell [9]. Despite similarities in their structure and infection mechanisms, IAV and IBV differ in many other ways. For instance, both IAV and IBV possess the hemagglutinin (HA) glycoprotein on their viral surfaces, which facilitates cell adsorption by binding to host sialoglycans during infection. However, the sialoglycan recognition properties of IAV and IBV significantly differ [10,11]. The HA of IAV can recognize both α2-3-linked and α2-6-linked sialic acids; avian IAV preferentially binds to α2-3-linked sialic acids, while human IAV and IBV preferentially bind to the α2-6-linked type. Some IBV strains may utilize additional receptors [12]. Additionally, IAV encodes PB1-F2 and PA-X, which are not encoded by IBV [13]. IBV uniquely encodes the NB protein, which has no counterpart in IAV [14]. Furthermore, IAV encodes the M2 ion channel protein, while IBV encodes a functionally similar but amantadine-insensitive BM2 protein [15]. The NS1 proteins of IAV and IBV have distinct roles in evading the innate immune response [16]. These biological differences between IAV and IBV contribute to their distinct host ranges and different levels of host adaptability.

Influenza virus assembly is a complex process that involves numerous cellular compartments. For example, after viral mRNA replication occurs within the nucleus, HA and neuraminidase (NA) are synthesized in the cytoplasm, after which they are subsequently concentrated into lipid rafts on the host cell membrane for assembly and budding [17,18]. HA and NA proteins expressed on this surface then interact with various host factors, thereby playing essential roles in viral attachment, entry, and release [19,20].

Sulfatide, a major glycolipid that is widely expressed in tissues, has been localized to lipid rafts, the Golgi apparatus, and lysosomes [21,22,23]. This glycolipid plays an essential role in myelin formation, signal transduction, cellular homeostasis, and resistance to microbial infection [24,25,26,27,28]. In viral infections, sulfatide acts as a binding co-factor and controls the entry of viruses, including human immunodeficiency virus-1 (HIV-1), vaccinia virus, and human parainfluenza virus (hPIV) [24,29,30]. Moreover, we also recently reported that sulfatide interacts with the norovirus outer shell protein [31].

We previously demonstrated that sulfatide binds to IAV HA, where it acts as a host molecule regulating viral infection [32,33,34]. When present on the cell membrane, sulfatide promotes the replication of IAV by enhancing efficient virus particle formation. Given this role, sulfatide may be a potential target for novel influenza virus therapeutics. However, the role sulfatide plays in IBV infection remains poorly understood. The amino acid sequence homology of HA between IAV and IBV is only 18% [35], raising the question of whether the sulfatide-binding capacity is conserved between these two virus types. Investigating sulfatide’s potential interaction with IBV may clarify which mechanisms of sulfatide-mediated viral replication are conserved or divergent between IAV and IBV.

In this study, we assessed the effect of sulfatide on IBV infection. First, we used cells with modified sulfatide expression levels to demonstrate that sulfatide promotes IBV replication. Furthermore, we confirmed that IBV specifically binds to sulfatide and showed that HA is the key viral component responsible for this interaction. In addition, when cells were treated with an anti-sulfatide antibody or a sulfatide synthesis inhibitor, we observed that sulfatase significantly suppressed IBV replication. Taken together, these findings indicate that sulfatide is crucial for the mid-to-late stages of IBV replication. This research provides the first evidence of sulfatide’s involvement in IBV replication, and clarifies its broader implications for the molecular pathogenesis of different influenza viruses.

## 2. Materials and Methods

### 2.1. Cell Lines

A549 (human lung carcinoma cell line, ATCC CCL-185), HEK293T (human embryonic kidney cell line, ATCC CRL-3216), COS7 (African green monkey kidney fibroblast-like cell line, ATCC CRL-1651), MDCK (canine kidney cells, ATCC CCL-34), and SulCOS1 cells were cultured at 37 °C and 5% CO_2_. SulCOS1 cells, a cell line established in our laboratory based on COS7 cells [32], were engineered to overexpress sulfatide by introducing ceramide galactosyltransferase (CGT) and cerebroside sulfotransferase (CST) genes into COS7 cells. HEK293T, A549, COS7, and SulCOS1 cells were cultured in Dulbecco’s modified Eagle’s medium with 10% fetal bovine serum (FBS). MDCK cells were grown in minimum essential medium (MEM) containing 5% FBS.

### 2.2. Viruses

IBV (B/Lee/1940, B/Shizuoka/96/2013, and B/Shizuoka/95/2013) were propagated in MDCK cells in serum-free medium (SFM) containing 2 µg/mL acetyl trypsin. After harvesting, the collected viral supernatant was concentrated by centrifugation at 12,000× *g* for 2 h at 4 °C. Viral pellets were resuspended in PBS and stored at −80 °C until further use.

### 2.3. Antibody

Mouse anti-sulfatide (GS-5) and anti-Gb3Cer (TU-1) antibodies were obtained from the culture supernatants of hybridomas previously generated in our laboratory [36,37,38]. In addition, an Anti-Human Influenza A, B, Rabbit Polyclonal antibody was purchased from Takara Bio Inc. (Shiga, Japan). Monoclonal Anti-Influenza B virus NP and Horseradish peroxidase (HRP)-conjugated protein A were purchased from Sigma-Aldrich (St. Louis, MO, USA), while Alexa Fluor^TM^ 594 goat anti-rabbit and Alexa Fluor^TM^ 555 goat anti-mouse antibodies were purchased from Invitrogen (Waltham, MA, USA). Finally, Peroxidase-conjugated AffiniPure Goat Anti-Mouse IgG (H+L) was purchased from Jackson ImmunoResearch (West Grove, PA, USA) and Anti-Myc-tag mAb was purchased from MBL Life Science (Tokyo, Japan).

### 2.4. Lipids

Sulfatide and galactosylceramide (GalCer) were both purchased from Fujifilm Wako Pure Chemical Corporation (Osaka, Japan). Glucosylceramide (GlcCer) was purchased from Nagara Science (Gifu, Japan). All glycolipids were dissolved in ethanol and stored at −20 °C.

### 2.5. Reagents

Sulfatase from *Aerobacter aerogenes* [39] was purchased from Sigma-Aldrich (St. Louis, MO, USA). UGT8 inhibitor 19 [40] and leptomycin B were purchased from Cayman Chemical (Ann Arbor, MI, USA) and dissolved in DMSO and an ethanol solution, respectively. U0126 was purchased from ChemScene (Monmouth Junction, NJ, USA) and dissolved in DMSO. All reagents were stored at −20 °C.

### 2.6. Plasmids

An IBV secreted hemagglutinin plasmid (IBV sHA) was constructed by cloning a modified version of the cDNA sequence of B/Lee/1940 hemagglutinin that excluded the transmembrane and cytoplasmic domains—to which a Myc-tag and a 6×His tag were added to the C-terminus. The resulting construct was then amplified by PCR from fragments synthesized as gBlocks (Integrated DNA Technologies, Coralville, IA, USA) and inserted into a pCAGGS vector using an NEBuilder HiFi DNA Assembly kit (New England Biolabs, Ipswich, MA, USA).

### 2.7. 50% Tissue Culture Infectious Dose (TCID_50_) Assay

Cells were seeded onto 96-well plates at a density of 2.0 × 10⁴ cells/well before being incubated overnight at 37 °C. Virus samples were serially diluted (1:10) from the original stock, and 100 µL of each dilution was used to infect the seeded cells. After 24 h of incubation at 37 °C, the cells were washed with PBS and fixed with methanol. Infected cells were labeled with Anti-Human Influenza A, B, Rabbit, Polyclonal antibody, followed by HRP-conjugated Protein A. Detection was performed using DEPDA buffer prepared using 0.1 M citrate buffer (pH 6.0) (10 mL), 30% H_2_O_2_ (1 µL), 0.06 M *N,N*-diethyl-*p*-phenylenediamine sulfate in acrylonitrile (200 µL), and 0.1 M 4-chloro-1-naphthol in acetonitrile (200 µL). All TCID_50_ values were calculated using the Kärber formula [41]. For comparison of virus infectivity between COS7, SulCOS1, and MDCK cells, equal viral loads were used for infection, and TCID_50_ values were measured for each cell type. For experiments examining the effects of GS-5 and sulfatase on infectivity, cells were treated with GS-5 for 1 h or sulfatase (0.1–0.2 units/mL) for 12 h prior to virus infection.

### 2.8. Measurement of IBV Replication

To compare viral replication in COS7, SulCOS1, and MDCK cells, the cells were seeded onto 48-well plates at a density of 5.0 × 10⁴ cells/well and incubated overnight at 37 °C. IBV was added at an MOI of 0.01 to each cell and cultured in SFM without acetyl trypsin at 37 °C for 24 h. The supernatants were then harvested, centrifuged at 800× *g* for five minutes at 4 °C, and stored at −80 °C. For proteolytic activation of HA, acetyl trypsin was added to the supernatant and incubated at 37 °C for 30 minutes. The titer of the virus sample was measured using the TCID_50_ assay. For GS-5 or TU-1 treatment, IBV was added to the cells at an MOI of 1 and cultured in the presence of GS-5 or TU-1 at 37 °C for 24 h. For sulfatase treatment, IBV-infected cells (MOI = 1) were cultured in SFM containing sulfatase (0.1~0.2 units/mL) before being incubated at 37 °C for 24 h. Subsequently, IBV was added to the cells in SFM containing sulfatase and incubated at 37 °C for 24 h. For UGT8 inhibitor 19 treatment, A549 cells were seeded onto 24-well plates and incubated with SFM containing UGT8 inhibitor 19 (200 nM) at 37 °C for 48 h prior to infection. Subsequently, IBV was added to the cells at an MOI of 1 in the presence of UGT8 inhibitor 19 in SFM and incubated at 37 °C for one hour. After the medium was removed, fresh SFM containing UGT8 inhibitor 19 was added, and the cells were incubated at 37 °C for another 24 h. Finally, the viral titer was assessed using focus-forming assays, as described below.

### 2.9. Focus-Forming Assays

MDCK cells were seeded in 12-well plates at a density of 2.0 × 10⁵ cells/well before being incubated overnight at 37 °C. Thawed virus samples were serially diluted and added to MDCK cells at 10 µL/well in SFM containing 1 µg/mL of acetylated trypsin. After 24 h of incubation, the infected cells were treated with DEPDA buffer, as per the protocol described above.

### 2.10. Enzyme-Linked Immunosorbent Assay (ELISA) for Sulfatide Binding to IBV

Sulfatide was immobilized on PolySorp immunoassay plates (Nargen Nunc International Japan, Tokyo, Japan) before being blocked with 0.5% defatted BSA at 4 °C for three days. After washing the plates five times with PBS, 2^6^ HAU of IBV in 0.05% defatted BSA-PBS was added, and the plates were incubated at 4 °C for two hours. After five additional washes with PBS, all wells were incubated with Anti-Human Influenza A, B, Rabbit, Polyclonal antibody diluted in 0.05% defatted BSA-PBS at 4 °C for another two hours. After five additional washes with PBS, the wells were incubated with HRP-conjugated Protein A at 4 °C for two hours. The wells were washed again five times with PBS, and the colorimetric reaction was initiated using *o*-phenylenediamine (OPD) in 0.1 M citrate buffer (pH 5.0). This was terminated after ten minutes by adding 1 N H_2_SO_4_. IBV binding was assessed by measuring the absorbance of the solution at a wavelength of 492 nm (control: 630 nm).

### 2.11. Thin-Layer-Chromatography (TLC) Overlay Assay

For TLC assays, sulfatide or glucosylceramide (3 nmol per lane) was spotted onto POLYGRAM^®^ plates (Macherey-Nagel, Düren, Germany) before TLC plates were developed using a developing solvent [Chloroform/Methanol/12 mM MgCl_2_ in water = 5/4/1 (*v*/*v*/*v*)]. After drying, the plates were blocked using 0.1% ovalbumin-1% polyvinyl pyrrolidone (pvp) in PBS at room temperature for two hours. After three washes with PBS, 2^8^ HAU/mL B/Lee/1940 with 0.1% ovalbumin-1% polyvinyl pyrrolidone in PBS was added. The plates were then incubated overnight at 4 °C. After three additional washes with PBS, all plates were incubated with Anti-Human Influenza A, B, Rabbit, Polyclonal antibody and HRP-conjugated Protein A 4 °C for two hours. IBV-binding glycolipids were detected using the DEPDA reagent according to the protocol described above, while glycolipids present on the plates were detected using an orcinol-sulfuric acid reagent.

### 2.12. Expression and Purification of IBV sHA

HEK293T cells were seeded in 10 cm cell culture dishes at a density of 2.0 × 10^6^ cells/dish and incubated overnight at 37 °C. IBV sHA plasmids were then transfected at 5 µg/dish into cultured cells using polyethylenimine (PEI) Max (Polyscience, Warrington, PA, USA) with Opti-MEM (Thermo Fisher Scientific, Waltham, MA, USA). After 72 h of incubation, the supernatant was collected and centrifuged at 800× *g* for five minutes at 4 °C before being filtered through a 0.45 µm filter. After adding 500 mM NaCl and 10 mM imidazole, the supernatant was subjected to His-tagged protein purification using Ni-NTA agarose beads (Fujifilm-WAKO). The purification of IBV sHA was detected by Western blots using anti-Myc-tag mAbs, followed by Peroxidase-conjugated AffiniPure Goat Anti-Mouse IgG (H+L). Finally, the protein concentrations were quantified by Coomassie Brilliant Blue (CBB) staining. CBB staining density was then measured by NIH ImageJ Fiji using a macro “Quantification of Gel Bands” [42].

### 2.13. Hemagglutination Assays

For the hemagglutination assays, guinea pig erythrocytes were purchased from Cosmo Bio (Tokyo, Japan) and washed three times with PBS. Purified IBV sHA was serially twofold diluted, then mixed with an equal volume of 0.5% normal-erythrocytes diluted in PBS. The mixture was incubated in 96-well clear round-bottom plates (Corning, Corning, NY, USA) at 4 °C for two hours, after which erythrocyte agglutination was observed.

### 2.14. ELISA Evaluation of HA-Glycolipid Binding

The binding assay for sHA and glycolipids was performed following the same ELISA protocol described above, with the following modifications: sHA samples of several concentrations were used instead of IBV, and sHA-glycolipid binding was detected using anti-Myc-tag mAb and Peroxidase-conjugated AffiniPure Goat Anti-Mouse IgG (H+L). Heat-denatured sHA at 95 °C for 10 minutes was used as a negative control.

### 2.15. Fluorescence Microscopic Imaging of IBV Infectivity Under Inhibited Sulfatide Synthesis

A549 cells were first seeded in 48-well plates at a density of 5.0 × 10⁴ cells/well and incubated overnight at 37 °C. Before infection, the cells were incubated with MEM containing UGT8 inhibitor 19 (200 nM) at 37 °C for 48 h. Subsequently, IBV was added to the cells in the presence of UGT8 inhibitor 19 in SFM and incubated at 37 °C for one hour. After the medium was removed, fresh SFM without acetyl trypsin containing UGT8 inhibitor 19 was added, and the cells were further incubated at 37 °C for 24 h. Following methanol fixation, infected cells were stained using an Anti-Human Influenza A, B, Rabbit, Polyclonal antibody, followed by an Alexa Fluor^TM^ 594 goat anti-rabbit antibody. IBV infectivity was then assessed by counting the number of infected cells per unit area.

### 2.16. IBV Replication When Sulfatide Synthesis Was Inhibited

A549 cells were first seeded in 24-well plates at a density of 5.0 × 10⁴ cells/well before being incubated overnight at 37 °C. Before infection, the cells were incubated with SFM containing UGT8 inhibitor 19 (200 nM) at 37 °C for 48 h. Subsequently, IBV was added to the cells at an MOI of 1 in the presence of UGT8 inhibitor 19 in SFM and incubated at 37 °C for one hour. After the medium was removed, fresh SFM containing UGT8 inhibitor 19 was added, and cells were incubated at 37 °C for another 24 h. Subsequently, the supernatants were harvested and centrifuged at 800× *g* for five minutes at 4 °C. The resulting clarified supernatant was then collected and stored at −80 °C until further use. Virus titer was assessed by a focus-forming assay.

### 2.17. Fluorescence Microscopy Imaging of Nuclear Export of the vRNP Complex

A549 cells were first treated with UGT8 inhibitor 19 (200 nM) at 37 °C for 48 h. Both treated and untreated cells were then seeded into CELLview Cell Culture Slide 10 Wells (Greiner, Kremsmünster, Austria) at a density of 2.5 × 10^4^ cells/well and cultured for nine hours. After washing with PBS, the cells were infected with IBV at an MOI of 3 and incubated at 37 °C for one hour in SFM. After another PBS wash, the cells were incubated at 37 °C for another eight hours in SFM containing U0126 (10 µM), UGT8 inhibitor 19 (200 nM), or leptomycin B (20 µM). After methanol fixation, fluorescence microscopy imaging was performed as described above using Monoclonal Anti-IBV NP and Alexa Fluor^TM^ 594 goat anti-rabbit, and DAPI solution.

### 2.18. Statistical Analyses

All statistical analyses were performed using the Student’s *t*-test as implemented in RStudio (Version: 2024.12.1+563) (R Core Team, Vienna, Austria).

## 3. Results

### 3.1. Sulfatide Promotes Viral Entry and Replication

To investigate the role sulfatide plays during IBV infection, we used transgenic cells that contain increasing amounts of sulfatide. Sulfatide is synthesized from ceramide via the sequential action of two enzymes: ceramide galactosyltransferase (CGT) and cerebroside sulfotransferase (CST) (Figure 1A). In a previous study, we established sulfatide-enriched cells, designated as SulCOS1 cells, by introducing CGT and CST into sulfatide-deficient COS7 cells [32]. COS7 cells are unable to synthesize sulfatide due to a deficiency in the CGT gene. In contrast, SulCOS1 cells overexpressing CGT and CST demonstrated a significantly increased sulfatide expression (Appendix A).

Next, COS7 and SulCOS1 cells were infected with the IBV strain B/Lee/1940 (hereafter referred to as B/Lee). We found that the infectivity of IBV in SulCOS1 cells was significantly higher than that in COS7 cells (Figure 1B). Moreover, the infectivity and replication of IBV in SulCOS1 cells were comparable to those observed in MDCK cells, which are widely used in influenza virus experiments and naturally express sulfatide (Appendix A). Compared to MDCK cells, COS7 cells exhibited lower viral infectivity and replication efficiency. Similar to IAV, IBV replication was higher in SulCOS1 cells than in COS7 cells (Figure 1C). To eliminate the influence of differences in infection susceptibility between COS7 and SulCOS1 cells, virus replication was compared by adding virus to achieve an MOI of 0.01 for each cell line, based on TCID_50_ calculations. Under these conditions, we observed enhanced IBV replication in SulCOS1 cells relative to COS7 cells; moreover, this effect was independent of the changes in infectivity. Taken together, these findings suggest that sulfatide promotes viral replication and transmission in IBV infection, as it does in IAV infection.

### 3.2. IBV Hemagglutinin Specifically Binds to Sulfatide

In a previous study, we demonstrated that IAV specifically binds to sulfatide, which is involved in the promotion of viral replication. To investigate whether IBV exhibits similar binding behavior with respect to sulfatide, we investigated the interactions between various IBV strains (B/Lee, B/Shizuoka/95/2013, and B/Shizuoka/96/2013) and sulfatide using Enzyme-Linked Immunosorbent Assays (ELISAs). Of these strains, B/Shizuoka/95/2013 belongs to the Yamagata Lineage, and B/Shizuoka/96/2013 belongs to the Victoria Lineage. We found that all the IBV strains tested bound to sulfatide (Figure 2A). We then performed subsequent analyses of the binding interaction between IBV and sulfatide. Specifically, we verified binding specificity using a thin-layer chromatography (TLC) overlay assay with sulfatide and glucosylceramide (Figure 2B). We found that IBV specifically bound to the position corresponding to sulfatide but not to the glucosylceramide position (Figure 2C).

Next, to further analyze the molecular mechanisms involved in the binding between IBV and sulfatide, we investigated which viral components interact with sulfatide. In IAV, hemagglutinin (HA), a type 1 transmembrane protein, is the specific viral component that interacts with sulfatide. Therefore, to determine whether HA in IBV has a similar binding property to sulfatide, we constructed an expression plasmid encoding secreted IBV HA (IBV sHA). Previous studies have shown that when the extracellular domain of HA is expressed, the recombinant protein is secreted into the extracellular space instead of being anchored to the membrane [43]. Given this information, we designed a construct that lacks the transmembrane and cytoplasmic domain of HA, but contains C-terminal Myc and His tags to facilitate detection and purification (Figure 3A). We found that IBV sHA was expressed in HEK293T cells, and protein secretion was confirmed by Western blot and Coomassie Brilliant Blue (CBB) staining (Figure 3B). Furthermore, IBV sHA can bind to sialic acid, as demonstrated by a hemagglutination assay (Figure 3C). Our ELISA results showed that IBV sHA can bind to sulfatide and that this binding activity is lost following heating to 95 °C for 10 min (Figure 3D). Moreover, IBV sHA showed lower binding abilities to galactosylceramide and glucosylceramide than to sulfatide (Figure 3E). Overall, these results suggest that HA is a viral component that may play a crucial role in mediating IBV-sulfatide binding.

### 3.3. Sulfatide Inhibition via Anti-Sulfatide Antibody or Sulfatase Suppresses IBV Replication

Sulfatide binds to HA, which emphasizes that interactions specifically occurring on the plasma membrane may play a critical role in regulating virus replication. Thus, to further investigate the effect of sulfatide on IBV infection, we investigated whether anti-sulfatide antibodies or a sulfatide-degrading enzyme resulted in inhibition of IBV replication. First, to evaluate the inhibitory effect of GS-5, an anti-sulfatide antibody, on IBV infection, MDCK cells were either pretreated with the antibody prior to infection or were cotreated with the antibody simultaneously during IBV infection. TU-1, an anti-Gb3Cer antibody [36], was used as a negative control to assess the specificity.

We found that pretreatment with GS-5 did not affect the initial infection of IBV in MDCK cells. However, GS-5 treatment significantly suppressed IBV replication compared to the untreated control (Figure 4A,B). In contrast, TU-1 treatment did not affect viral replication. Interestingly, we observed a similar suppressive effect on IBV replication when cells were treated with sulfatase, an enzyme that removes sulfate groups from sulfated glycoconjugates, including sulfatide (Figure 4C,D), with no cytotoxicity at the tested concentration (Appendix A). Since GS-5 and sulfatase likely do not pass through the cell membrane, these results suggest that sulfatides expressed on the plasma membrane are critical for regulating IBV replication.

### 3.4. Inhibition of Sulfatide Synthesis Suppresses IBV Replication and Viral Ribonucleoprotein Translocation in Human Lung Derived-Cells

Next, we investigated the effects of sulfatide reduction on initial IBV infection and IBV replication in human alveolar basal epithelial A549 cells. As described above, sulfatide biosynthesis requires two key enzymes: CGT and CST (Figure 1A). To inhibit sulfatide synthesis, A549 cells were treated with UGT8 inhibitor 19, a small molecule that targets CGT and suppresses the synthesis of galactosylceramide, which is a precursor of sulfatide. We found that while treatment with the UGT8 inhibitor 19 did not affect initial IBV infection, it significantly reduced viral replication (Figure 5A,B). Cytotoxicity assays confirmed that UGT8 inhibitor 19 showed no cytotoxicity at the concentration used (Appendix A). Taken together, these results suggest that sulfatide plays a critical role during IBV replication.

To further explore how sulfatide regulates IBV replication, we assessed the nuclear export of viral ribonucleoprotein (vRNP). vRNP is a complex that contains genomic RNA, viral polymerase, and nucleoprotein (NP). During influenza virus infection, vRNP is replicated in the nucleus, transported out of the nucleus, and eventually incorporated into progeny viral particles [8,44]. A previous study from our lab showed that sulfatide expressed on the cell membrane enhances IAV replication by promoting vRNP nuclear export. Thus, to determine whether this mechanism also occurs with IBV, we evaluated the impact of cellular sulfatide reduction on vRNP nuclear export in IBV-infected A549 cells treated with UGT8 inhibitor 19. We also used two positive controls: leptomycin B, a specific inhibitor of the nuclear export factor CRM1, and U0126, an MEK inhibitor known to suppress vRNP nuclear export. As expected, cells treated with U0126 exhibited vRNP retention within the nucleus, while in untreated cells, vRNP fluorescence leaked out of the nucleus and into the cytoplasm (Figure 5C). Moreover, cells treated with the UGT8 inhibitor 19 or leptomycin B exhibited vRNP retention within the nucleus (Figure 5D), despite—once again—untreated cells showing vRNP fluorescence that leaked out of the nucleus and into the cytoplasm (Figure 5D). Taken together, these findings suggest that sulfatide facilitates vRNP nuclear export and thereby regulating viral replication during IBV infection.

## 4. Discussion

In a previous study, we found that sulfatide promotes viral replication in IAV-infected cells [32]. However, whether sulfatide had a similar effect on other types of influenza virus infections, including IBV, remains unclear. Some influenza drugs, such as NA inhibitors, are known to be effective against both IAV and IBV, whereas others, such as the M2 inhibitors amantadine and rimantadine, target only IAV. It is therefore important to understand the roles that sulfatide plays during both IAV and IBV infections; moreover, this may be crucial for developing sulfatide-based therapeutic strategies. In this study, we demonstrated that sulfatide specifically binds to the HA of IBV and promotes IBV replication. We also found that treatment with an anti-sulfatide antibody, sulfatase, or an inhibitor of sulfatide biosynthesis significantly suppressed IBV replication. Furthermore, inhibitors of sulfatide biosynthesis were also associated with a decrease in vRNP nuclear export, suggesting that sulfatide-induced enhancement of viral replication is linked to higher rates of vRNP nuclear export. These findings emphasize the critical role sulfatide plays during IBV replication and offer new insights into its interactions with host factors. The proposed mechanism of sulfatide’s role during IBV infection is depicted in Figure 6. Briefly, sulfatide interactions within lipid rafts likely facilitate HA-mediated vRNP nuclear export, thereby promoting viral replication during the middle to late stages of infection.

Subsequently, a comparative analysis of COS7 and SulCOS1 cells revealed that sulfatide enhances IBV replication, thereby supporting its role as a key co-factor during IBV infection. However, unlike previous observations of IAV [32], IBV showed differences in both infectivity and replication between these cells, possibly due to differences in receptor binding specificity (Figure 1B). For example, IBV predominantly binds to α2-6-linked sialic acids, while IAV exhibits a broader range of receptor specificities [10,11]. Moreover, flow cytometry analysis revealed an increase in α2-6-linked sialic acid on the plasma membrane surface of SulCOS1 cells, suggesting that there are important differences in glycan structures on the plasma membrane. Thus, in SulCOS1 cells, prolonged sulfatide overexpression may have altered the cellular glycan composition. In contrast, transient sulfatide reduction in A549 cells affected IBV replication but not its infectivity. This suggests that transient modulation of sulfatide levels has a minimal impact on glycan composition. In contrast, a significant correlation between sulfatide levels and viral replication efficiency was observed in both COS7 and A549 cells. Overall, these results therefore suggest that sulfatide contributes to IBV replication rather than to one of the initial stages of infection.

Another important observation was that the IBV HA specifically binds to sulfatide, as demonstrated by ELISA. Further analyses using recombinant HA constructs lacking transmembrane and cytoplasmic domains revealed that this binding affinity was higher than that of other structurally similar glycolipids, including galactosylceramide and glucosylceramide. Interestingly, HA bound to sulfatide only via its extracellular domain, suggesting that IBV HA binds to the 3-*O*-sulfated galactose structure of sulfatide on the plasma membrane. Similar results have been obtained using recombinant IAV HA proteins [34,45], suggesting that the HA extracellular domain-sulfatide interaction mechanism is conserved between IAV and IBV.

Furthermore, UGT8 inhibitor experiments in A549 cells demonstrated that the inhibition of sulfatide biosynthesis reduced IBV replication. Previous studies using canine MDCK cells and monkey COS7 cells have also found that sulfatide plays a role during influenza virus infection [32]. However, this report is the first to demonstrate that the role of sulfatide in promoting influenza virus replication is conserved in human cells. Moreover, the suppression of vRNP nuclear export observed in response to the inhibition of sulfatide synthesis using a UGT8 inhibitor revealed the mechanistic basis of sulfatide-mediated IBV replication enhancement. Since similar suppression of vRNP nuclear export has been observed in IAV-infected cells [32], these results suggest that the function of sulfatide is conserved across different influenza virus types. To clarify how the inhibition of sulfatide synthesis causes the suppression of vRNP nuclear export and decreases viral replication, future studies are necessary including Western blot for viral protein expression and qRT-PCR for viral RNA levels. As previously reported [46], we also confirmed that the nuclear export of vRNPs was inhibited by the MEK inhibitor U0126 (Figure 5C). Further analysis, including examination of the localization of viral proteins such as HA, is needed to clarify how sulfatide affects the regulation of vRNP nuclear export and IBV replication. In an earlier study, Marjuki et al. demonstrated that activation of the MAPK/ERK pathway during influenza virus infection was facilitated by the accumulation of HA in lipid rafts, which subsequently promoted vRNP nuclear export [47]. In general, lipid rafts act as regulatory hubs that modulate signaling pathways [48]. Sulfatides therefore likely modulate the MAPK/ERK pathway via interactions with HA within lipid raft domains, which subsequently leads to enhanced rates of viral replication.

Overall, this study demonstrated that exogenous treatments, such as sulfatide biosynthesis inhibitors or sulfatase enzymes, can effectively inhibit influenza virus replication. These findings enhance our general understanding of IBV-host interactions and position sulfatide as a promising target for future antiviral strategies. The specific interactions between sulfatide and HA, along with the accessibility of extracellular sulfatides, make them particularly attractive targets for future therapeutic interventions. Moreover, the conserved role of sulfatides in regulating viral replication across different influenza strains suggests the potential for developing broadly effective therapies.

Future studies should focus on elucidating the structural basis of sulfatide-HA interactions and exploring the impact of sulfatide inhibition using in vivo models. Interactions between sulfatide and viral proteins other than HA will need to be analyzed to demonstrate that the HA-sulfatide interaction specifically enhances IBV replication. Furthermore, the potential involvement of other host factors in lipid rafts and their roles they may play in ERK activation by sulfatide and HA should be investigated. Notably, the inhibition of sulfatide biosynthesis may alter the composition of other glycosphingolipids, including precursors such as galactosylceramide. Therefore, a multifaceted experimental approach is required to fully elucidate the functional role of sulfatide and to map how it interacts with host factors during influenza infection.

In conclusion, this study demonstrates that sulfatide-HA interactions on the plasma membrane promote IBV replication by facilitating vRNP nuclear export (Figure 6). Furthermore, our findings suggest that sulfatide present within lipid rafts interacts with newly synthesized HA during the middle to late stages of infection, thereby causing enhanced viral replication by making vRNP nuclear export more efficient. These insights not only establish a foundation for developing sulfatide-targeting antiviral therapies but also highlight the critical role of glycosphingolipids, including sulfatide, in host-virus interactions, thereby advancing our understanding of glycan-mediated influenza pathogenesis.

## Figures and Tables

**Figure 1 viruses-17-00530-f001:**
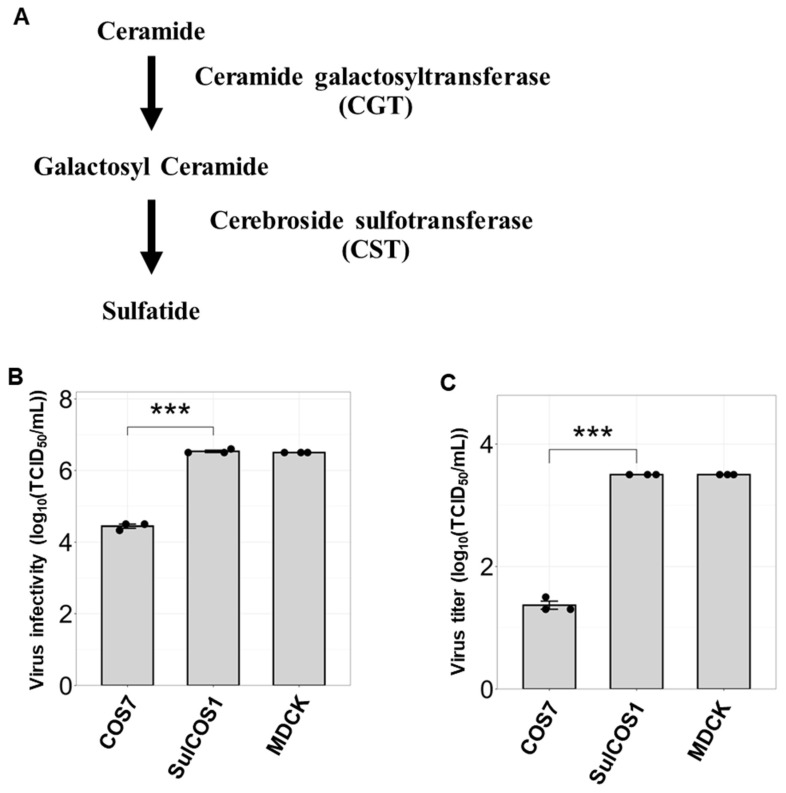
Sulfatide enrichment enhances influenza B virus infection and its replication. (**A**) Sulfatide biosynthetic pathway. (**B**) Virus infectivity was quantified using the TCID_50_ assay. IBV B/Lee/1940 strain was used to infect COS7 cells, sulfatide-expressing COS7 cells (SulCOS1), and MDCK cells at equal viral loads. SulCOS1 cells were generated by the stable expression of ceramide sulfotransferase (CST) and ceramide galactosyltransferase (CGT) genes in sulfatide-deficient COS7 cells. MDCK cells were used as the positive control. (**C**) Viral replication was assessed using TCID_50_ assays. Cells were infected with IBV at an MOI of 0.01, and the viral titer in the culture supernatants was determined 24 h post-infection. Data were obtained from three independent experiments. Values are presented as the mean ± SD. *** *p* < 0.001.

**Figure 2 viruses-17-00530-f002:**
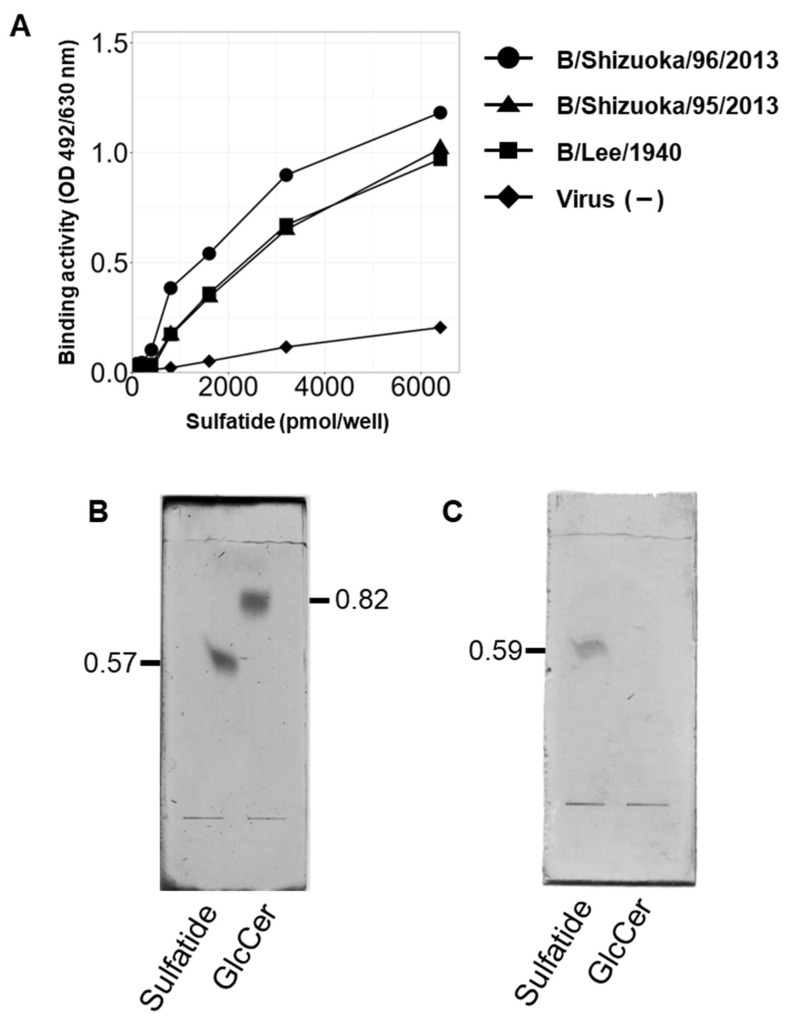
IBV exhibits binding affinity to sulfatide. (**A**) Analysis of IBV binding to sulfatide using enzyme-linked immunosorbent assay (ELISA). Three IBV strains (B/Lee/1940, B/Shizuoka/96/2013, and B/Shizuoka/95/2013) were added at 2^6^ HAU/well to sulfatide-coated plates, and the binding activity was determined using HRP-conjugated antibodies. Data were obtained from three independent experiments. The values indicate the mean ± S.D. (**B**,**C**) Thin-layer chromatography (TLC) overlay assay of IBV glycolipid binding. (**B**) Glycolipids were visualized using orcinol sulfate reagent. (**C**) IBV (B/Lee/1940) binding to glycolipids was detected using anti-influenza A and B rabbit polyclonal antibodies and HRP-conjugated anti-rabbit IgG. The values indicated in (**B**,**C**) are the retention factor (Rf) values.

**Figure 3 viruses-17-00530-f003:**
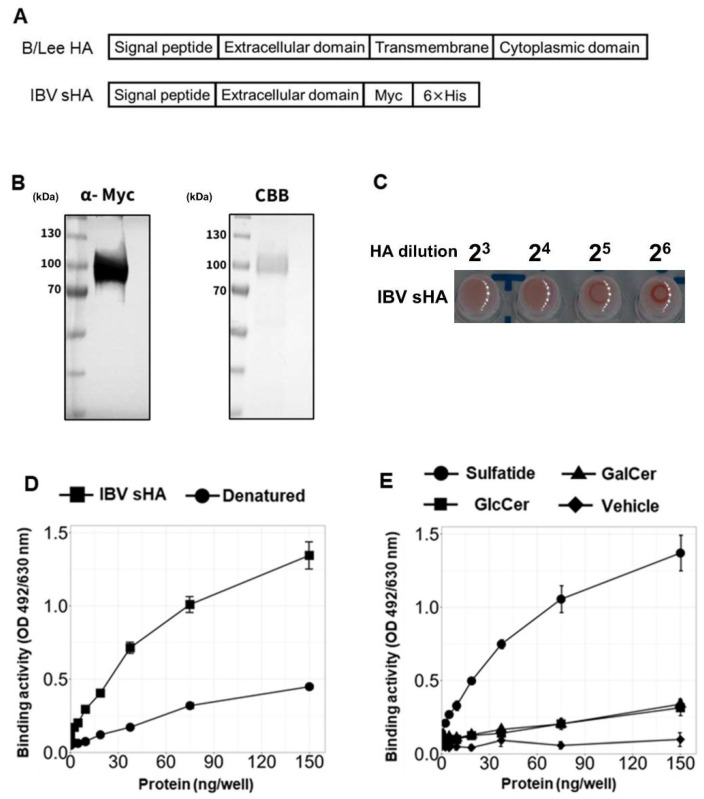
Soluble IBV HA binds to sulfatide. (**A**) Schematic representation of the secreted IBV HA (IBV sHA) constructs. This construct was generated by removing the transmembrane and cytoplasmic domains of B/Lee/1940 HA and then inserting the resulting sequence into the pCAGGS vector with C-terminal 6×His and Myc tags for detection and purification. (**B**) Detection of IBV sHA in culture supernatants via Western blotting using an anti-Myc antibody (left) and Coomassie Brilliant Blue (CBB) staining (right). (**C**) Hemagglutination activity of purified IBV sHA (85 ng/µL) assessed by serial twofold dilution with 0.5% guinea pig erythrocytes in PBS. (**D**,**E**) Analysis of IBV sHA binding to sulfatide as assessed using ELISA. For this test, sulfatides or other glycolipids were immobilized at a concentration of 6.4 nmol/well. (**D**) Effect of heat denaturation (95 °C, 10 min) on IBV sHA binding to sulfatide. (**E**) Binding specificity of IBV sHA, as determined by a comparative analysis of various glycolipids. The data are from three independent experiments for (**D**,**E**). The values are presented as the mean ± SD.

**Figure 4 viruses-17-00530-f004:**
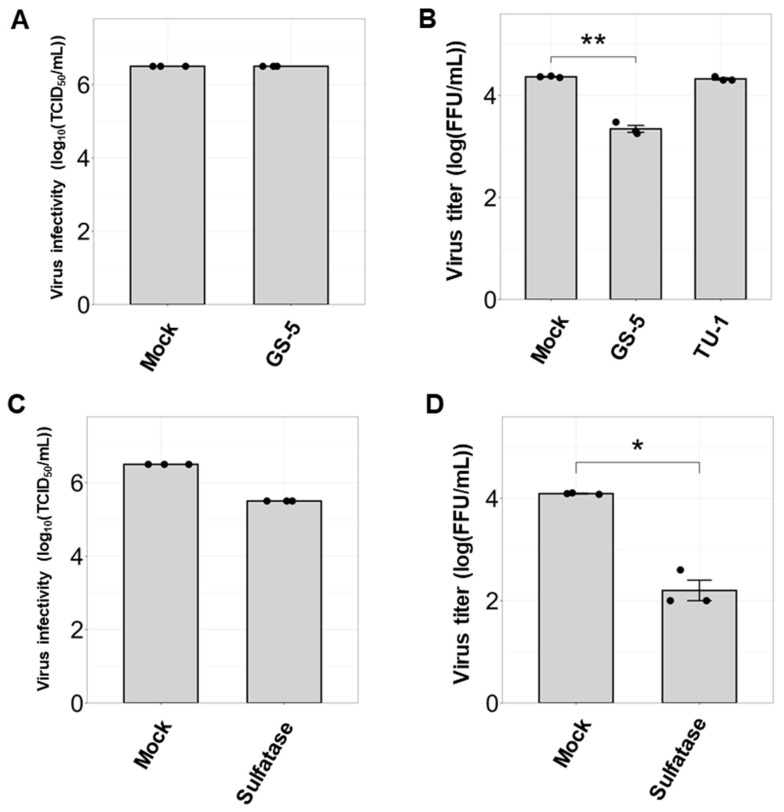
Sulfatide inhibition suppresses IBV replication. (**A**) Analysis of B/Lee/1940 infectivity in MDCK cells following sulfatide inhibition. Cells were pretreated with the anti-sulfatide monoclonal antibody GS-5 prior to infection. TCID_50_ was determined 24 h post-infection. (**B**) Quantification of IBV replication under sulfatide inhibition, as determined using a focus-forming assay. MDCK cells were treated with GS-5 during infection with B/Lee/1940 (MOI = 1), and anti-Gb3Cer monoclonal antibody TU-1 was used as a negative control. The virus titer of the culture supernatants was determined 24 h post-infection. (**C**) Effects of sulfatase-mediated sulfatide degradation on virus infectivity. Cells were pretreated with sulfatase prior to infection. (**D**) Impact of sulfatase treatment on IBV replication. Here, sulfatase was added simultaneously with virus infection. Data were obtained from three independent experiments. Values are presented as the mean ± SD. * *p* < 0.05, ** *p* < 0.01.

**Figure 5 viruses-17-00530-f005:**
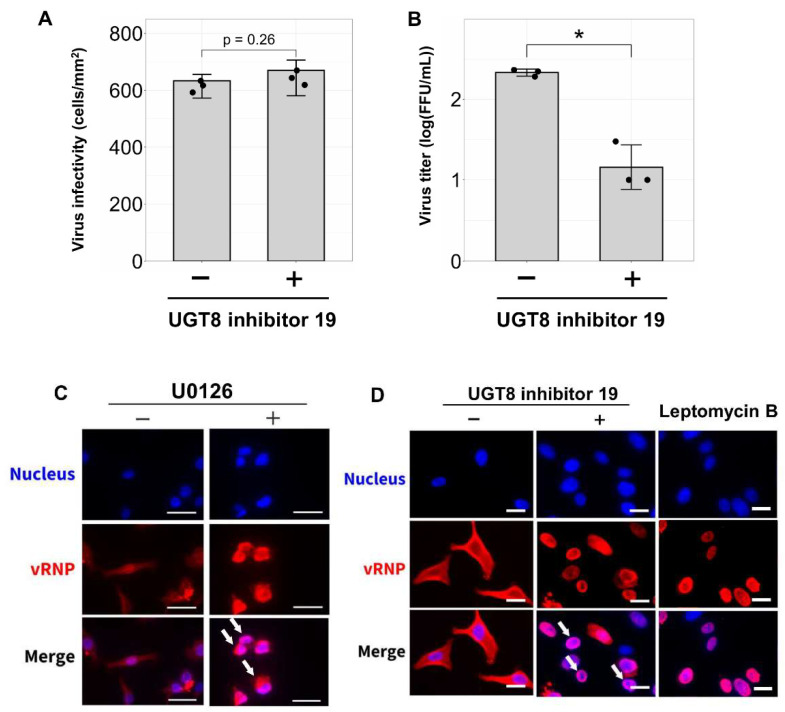
Inhibition of sulfatide biosynthesis impairs IBV replication and viral ribonucleoprotein transport. UGT8 inhibitor 19, an inhibitor of galactosylceramide synthase, was used to inhibit sulfatide biosynthesis. Cells were pretreated with 200 nM UGT8 inhibitor 19 for 72 h prior to infection. (**A**) Quantification of B/Lee/1940 infectivity in A549 cells following inhibition of sulfatide synthesis. Infected cells were counted 24 h post-infection and expressed as infected cells/mm^2^. (**B**) Analysis of IBV replication in A549 cells with inhibited sulfatide synthesis, as measured using focus-forming assays. (**C**,**D**) Immunofluorescence analysis of nuclear export of viral ribonucleoprotein (vRNP). A549 cells infected with B/Lee/1940 were immunostained for nuclei (DAPI, blue) and vRNP (anti-IBV NP followed by Alexa Fluor 555-conjugated anti-mouse IgG; red). White arrows indicate cells with impaired vRNP nuclear export. (**C**) Effects of the ERK signaling inhibitor U0126 on vRNP nuclear export. U0126 was used to confirm the role of ERK signaling in vRNP nuclear export. (**D**) Impact of UGT8 inhibitor and leptomycin B on vRNP nuclear export. Leptomycin B, a nuclear export inhibitor, was used as the positive control. The scale bar represents 40 µm. Data were obtained from three independent experiments for (**A**,**B**). Values are presented as the mean ± SD. * *p* < 0.05.

**Figure 6 viruses-17-00530-f006:**
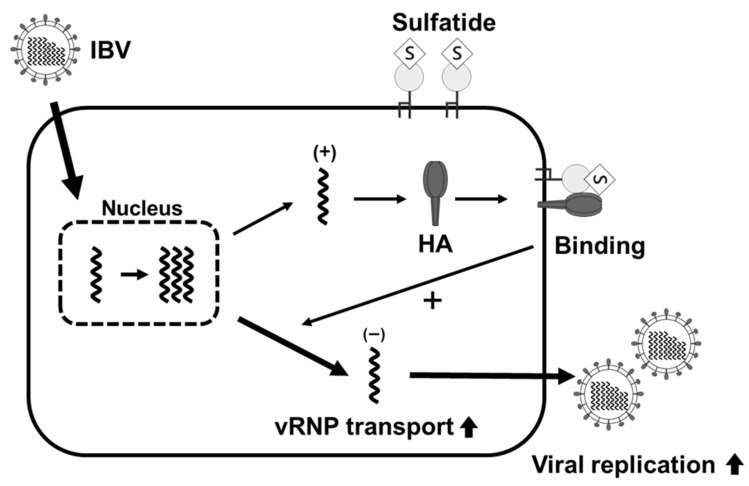
Predicted mechanism by which sulfatide promotes IBV replication.

## Data Availability

The raw data supporting the conclusions of this article will be made available by the authors on request.

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
