# Peer review of "Sulfatide Binds to Influenza B Virus and Enhances Viral Replication"

_viruses, 2025, doi:10.3390/v17040530_

Round 1
Reviewer 1 Report
Comments and Suggestions for Authors
Authors of the manuscript Sulfatide binds to influenza B virus and enhances viral replication investigate the influence of sulfatide on the replication of influenza B virus. Their results showed that sulfatide promotes replication of the influenza B virus and binds the HA. On the other hand, sulfatide inhibitors suppressed viral replication and impairs the export vRNP from the nuclei. The paper is not well written. Text is sometimes very confusing, many important informations are missing. To understand this manuscript, you should read previously published work with influenza a virus.
General Comments:
-Introduction is very vague. Authors try compare influenza a and influenza B viruses. They state that there are differences, but specific differences are not mention. For example: The avian IAVs bind SA α2,6. Human IAVs and IBVs preferentially bound to SAα2,3. Some IBVs may be able to utilize other receptors than SAα2,6 and SAα2,3. IAV encode about 18 proteins. Most of recently find proteins are not encode by IBA. IAV encode M2, thereby inhibitors against M2 cannot work with IBV. IBV encode BM2. IBV encode NB protein, which is not in IAV. Function of NS1 protein of IAV differs from NS1 of IBV. Replication is similar.
-You should mention that some NA inhibitors work against IBV.
-Can you provide % of homology of HA from IAV and IBV?
-Materials and methods- authors should describe better used cells. Source (ATCC and characteristic of the cells will really help. Describe properly Soc7, SulCos1 cells.
-Line 152-154. Authors claim: “then the luminescent reaction was initiated using O-phenylenediamine (OPD) in 0.1 M citrate buffer (pH 5.0). This was terminated after ten minutes by adding 1 N Hâ‚‚SOâ‚„. IBV binding was then assessed by measuring solution absorbance at 492 nm (control wavelength: 630 nm).“ They did not measure luminescence.
- Legend Fig 1. The text about COS7 and SulCOS1 is confusing. How differ these cell lines?
- Why authors used MDCK cells?
- Fig1B and Fig1C legend should be change. B is titration of the virus on the SOC7 and SulSOC1. I assume that the same amount of the virus was used. MDCK cells should have sense if you can compare MDCK ASA cells.
- Legend 2. Specify the virus. Was it B/Lee?
- Is focus forming assay plaque assay?
Reviewer 2 Report
Comments and Suggestions for Authors
This manuscript describes the impact of sulfatide on influenza B virus (IBV) infection. Sulfatide, a widely expressed glycolipid, has been previously shown to modulate influenza A infection. Here, the authors demonstrate that sulfatide can bind to the extracellular region of IBV hemagglutinin. Furthermore, they modulate sulfatide expression through multiple methods to determine the impact on IBV titer. They show that IBV vRNP nuclear export is impaired after treatment with an inhibitor of sulfatide biosynthesis, suggesting that the promotion of viral replication by sulfatide involves an interaction with HA and the nuclear export pathway.
General comments
- While the authors have shown an interaction between IBV HA and sulfatide, they have not provided evidence to rule out interactions with other viral proteins. The significance of the interaction between sulfatide and HA (shown in Fig 3) would be improved by including additional viral proteins (eg IBV NEP). It would also provide greater support to the proposed mechanism of action.
- The infectivity assay (described in 2.13) is designed to measure released virus. The impact of sulfatide inhibition on virus infectivity should also be measured at the early stages of infection, eg: assessment of newly synthesised viral protein expression during the first cycle of replication (8 to 16 hours) by flow cytometry and levels of viral RNA in infected cells and cell supernatants at different times by RT-PCR. Particularly given the interaction between sulfatide and HA, a more direct measurement of the impact on viral entry is needed.
- Similarly, the impact on viral replication could be demonstrated by looking at more than just released virus. Western blots for viral protein expression or qRT-PCR for viral RNA expression would improve the hypothesis that sulfatide inhibition only acts through modulation of the nuclear export pathway at the mid-late stages of replication.
- Do these treatments (sulfatase, UGT8 inhibitor 19) cause any cellular toxicity that would impact viral titres? It would be useful to include some toxicity data
Specific comments
- Materials and methods: There seems to be a bit of repetition in this section, could some of the subsections be combined?
- Fig 1B should include the relative expression levels of sulfatide in the cell lines to better contextualise these results.
- Line 276: Did you confirm that all cells were infected in all cell lines? It seems that the reduced infectivity in COS7 cells would still impact this result
- Line 342: Describes TU-1 control, but is not included in Fig 4A
- Fig 5: Does the localisation of other viral proteins change, or does treatment only affect vRNP nuclear export? It would be useful to include an additional viral protein, such as HA, as a control to see the impact of sulfatide inhibition on viral protein trafficking more generally.
Round 2
Reviewer 1 Report
Comments and Suggestions for Authors
I have no other comments.
Reviewer 2 Report
Comments and Suggestions for Authors
The revised manuscript by Kurebayashi et al is much improved. Many of the reviewer comments have been addressed, including the addition of cytotoxicity assays and confirmation of sulfatide expression levels. Other suggested experiments have been included in the discussion, plus additional justification of the study has been added to the introduction, which helps to contextualise their results.